# Effects of Personality and Behavioral Syndromes on Competition for Social Hierarchical Status in Anemonefish *Amphiprion clarkii*

**DOI:** 10.3390/ani14152216

**Published:** 2024-07-30

**Authors:** Lisheng Wu, Shunyun Deng, Wei Tang, Sipeng Zhang, Feng Liang, Shaoxiong Ding

**Affiliations:** State Key Laboratory of Marine Environment Science, College of Ocean and Earth Sciences, Xiamen University, Xiamen 361000, China; lisheng@xmu.edu.cn (L.W.); 22320201151272@stu.xmu.edu.cn (S.D.); tw6622@163.com (W.T.); jasper0517@163.com (S.Z.); maximum.liang@gmail.com (F.L.)

**Keywords:** anemonefish, *Amphiprion clarkii*, social hierarchy, personality, behavior

## Abstract

**Simple Summary:**

In marine coral reef ecosystems, anemonefish are a group with highly complex behavioral patterns and strict social hierarchies. However, there has been a paucity of research on the factors associated with stable hierarchical relationships or the behaviors performed during competitive interactions, especially in anemonefish in the growth phase before sexual differentiation. In the present study, we employed an observational approach to assess the personality type of each individual by observing the interactive behavior between unsexually differentiated *Amphiprion clarkii*. Our findings revealed that the personality of *A. clarkii* varied significantly between individuals, with two main types. Furthermore, our pairing experiments confirmed the impact of personality on the establishment of a stable social hierarchy. These personality-related behavioral traits are potentially important for both individual and population survival, and extend our understanding of the establishment and evolution of social hierarchies in anemonefish and their impact on population dispersal. Furthermore, they have the potential to guide aquaculture practices.

**Abstract:**

In this study, the behavioral ethogram of *Amphiprion clarkii* during the growth phase prior to sexual differentiation was summarized based on behavioral observations in three social environments. These behaviors can be classified into four categories: in addition to normal behaviors, the other three categories of behaviors—threatening, agonistic, and appeasing behaviors—represent different intentions in interactions with other individuals. Subsequently, the personalities of each individual were assessed by testing their reactions to intruders. These individuals mainly exhibited two distinct personality types: bold-aggressive and shy-submissive. In pairing experiments, the interactive behaviors of the anemonefish were observed in pairing combinations of different body sizes and personalities. The impact of personality on the establishment of a stable social hierarchy was confirmed by significant differences in the success rates of different pairing combinations, with the frequency of appeasing behaviors being the main factor influencing the success rate. Our results suggested that in natural waters, when juvenile individuals migrate among host anemones, shy-submissive individuals are more likely to be accepted due to their appeasing behaviors towards larger individuals, thus avoiding the risk of being attacked and bitten, and benefiting the survival of the individual. Conversely, bold-aggressive individuals are more likely to be driven away to another host anemone due to their unwillingness to settle for a lower-ranked status, thereby contributing to population dispersal and increasing opportunities for gene exchange between populations.

## 1. Introduction

Anemonefish (Pomacentridae: *Amphiprion*) inhabit reef areas and form symbiotic mutualisms with sea anemones [1]. In marine coral reef ecosystems, anemonefish comprise a group with highly complex behavioral patterns and strict social hierarchies. A breeding pair of a functional female and male occupies the dominant position in the group, with several sexually undifferentiated non-breeders of different sizes. The dominance rank within the group is established based on the relative body size, and this hierarchy can fluctuate if individuals move between groups [2]. It is commonly assumed that larger individuals are more aggressive (fighting force), and numerous studies have examined how body sizes correspond with the contest outcome, i.e., winning or losing [3,4,5]. However, there has been limited investigation of the factors associated with stable hierarchical relationships or the behaviors performed during competitive interactions.

Various behavioral interactions are commonly observed within and between anemonefish groups. Presently, studies on anemonefish behaviors have mainly focused on aspects involved in reproduction and sexual reversal, such as territorial protection, egg care, and dominance [2,6]. Nevertheless, for anemonefish in the growth stage before sexual differentiation, many aspects regarding the establishment of a social hierarchical structure, behavioral traits and personalities remain unclear. 

Animal personality is defined as a behavioral pattern that can describe multiple behavioral traits, including boldness, activity, aggression and sociability, and this study explores the relationship between these traits over time [7,8]. Fish with distinct personality types exhibit different behavioral syndromes in their life activities and responses to environmental stresses, as well as in their competition for hierarchical status within groups [9]. Certain behavioral traits tend to be correlated across individuals; for example, boldness was observed to be positively correlated with aggressiveness in *Thymallus thymallus* [10], *Salmo salar* [11], and *Salmo trutter* [12,13]. Bold individuals often accept higher risk in return for an increased foraging reward and growth rate, while shy individuals tend to make more cautious choices [8,14]. Anemonefish also exhibit a typical bold–shy trait and consistent behavioral traits that benefit them [15]. For instance, shy individuals of *Amphiprion percula* were observed to spend longer with their symbiotic sea anemone compared to bold ones, leading to increased food intake and gas exchange [6]. Aggressive behaviors are commonplace in the social interactions of fish, including competition for food resources and mates, territorial defense and larval protection [16]. To overcome the limitations of solely describing an animal’s personality, it is often necessary to assess multiple behavioral traits [7]. Behavioral assessment offers advantages in the study of animal personality, allowing for a more comprehensive understanding [17,18]. Therefore, describing and quantifying the behavioral traits of anemonefishes in various contexts could enhance our understanding not only of anemonefish behaviors and personality types, but also of animal behaviors in general [19]. 

*Amphiprion clarkii* (Bennett, 1830) is a broadly distributed anemonefish species, as well as the least host-specific anemonefish and the most interesting model species [20,21,22]. Meanwhile, owing to its typical behavioral traits and weak reliance on sea anemones, observing the behaviors of *A. clarkii* in laboratory settings is readily feasible [23].

The aim of the present study is to investigate whether the personality and behaviors of anemonefish influence the success of establishing social hierarchies. Before delving into this inquiry, it is imperative to identify the behavioral traits and ascertain whether distinct personality types exist among anemonefish. In addition to its body size, do the behavioral traits and personality of an individual impact its position within the social hierarchy? Moreover, what is the potential significance of these personality-related behavioral traits in anemonefish for both individual and population survival? These questions have garnered considerable interest due to their potential ability to guide aquiculture practices and expand our understanding of the establishment and development of the social hierarchy structure among anemonefish, as well as their effects on population dispersal. Furthermore, these findings can serve as a valuable reference in the behavioral and ecological studies of bony fishes and other vertebrates. In this study, the individual behavioral ethograms of *A. clarkii* prior to sexual differentiation were examined and summarized through extensive behavioral observations (May 2016 and March 2018). Subsequently, we determined the individual personality types of each anemonefish through personality-testing experiments. Following this, we investigated the influence of personality and body size on the success of establishing social hierarchies through the observation of behaviors in pairing experiments.

## 2. Materials and Methods

### 2.1. Animal Preparation

The experiments were conducted at Xiamen University, China. The *A. clarkii* individuals used in the experiments were from the Ornamental Fish Aquatic Farm in Hainan Province, China, with body lengths of 1.8–3.0 cm (2.4 ± 0.3 cm). Upon arrival at the laboratory, all fish were in an early growth stage, and they remained sexually undifferentiated throughout the entire experimental period. Polyvinyl chloride (PVC) tubes served as shelters and habitats in the experiments.

The *A. clarkii* individuals were fed commercial pellets (45.5% crude protein, 10.5% lipid, 9.5% moisture and 9.8% ash, purchased from Xiamen Jiakang Feed Co., Ltd., Xiamen, China) daily throughout the experiment, while artificial seawater with a salinity of 29–32 PSU was utilized for cultivation. The temperature was maintained within the range of 24 to 29 °C, with pH levels monitored and controlled between 8.0 and 8.2. Water nitrate and ammonia nitrogen were detected regularly. During the experiment, approximately 20–25% of the water was replaced each month, supplemented with nitrifying bacteria and pure water on a weekly basis. The entire aquaculture tank underwent routine cleaning to mitigate the accumulation of harmful substances in the water. Recordings of *A. clarkii* behaviors were captured using a Canon camera (EOS 70D, Canon Co., Tokyo, Japan) in recording mode.

### 2.2. Behavioural Ethogram Recording

Moyer and Bell [24] previously recorded and defined anemonefish behaviors in a study on the reproductive behaviors of mature *A. clarkii*. Building upon this, we conducted similar recordings and analyses of behaviors in sexually undifferentiated *A. clarkii*. To better observe the behaviors of *A. clarkii* in various social environments, the individuals were assigned into three context groups: a solitary group (with only one individual in the tank, and an adjacent compartment isolated with a black acrylic plate to prevent visual signal communication), a shared group (two individuals in the same tank) and a separated group (each individual isolated by glass, with communication only through visual signals). As there were no significant differences observed in the behaviors between morning and afternoon [15], behavioral recordings were conducted once daily. The behavioral recording for each individual lasted for 12 min, with the initial 2 min allocated to acclimation. The behavioral recordings of individuals in these 3 groups were manually observed, described and summarized in a behavioral ethogram.

### 2.3. Personality Assessment

According to the methodology of personality trait testing used in previous studies [25,26], two tests were used to assess the personality of each *A. clarkii* individual, specifically their reaction to two conspecific intruders. The first was an anemonefish model (of the same size as the examined individuals) introduced into the tank, and the second was a mirror image of itself (a mirror was placed in the tank). The latency time to first attack and the frequency of attacks on the model and mirror were observed and recorded within a 10 min period. To ensure the consistency of individuals’ personality, the behavioral responses of each individual were tested twice in the same context, with a 24 h interval between each test. That is, after completing the first round of personality assessment for each individual, a second round was conducted.

The behaviors of each individual and their frequencies were recorded in two tests and subsequently utilized to conduct clustering and discriminant analysis, aiming to assign a personality type (bold-aggressive or shy-submissive) to each individual. A total of 50 individuals underwent personality assessment, with 2 individuals accidentally injured and ineligible for the subsequent pairing experiments.

### 2.4. Pairing Experiment

Following the personality assessment, contests for hierarchical status were staged between pairs of non-breeding individuals within coexisting tanks, analogous to the natural contests that occur at the outset of group formation in the wild. The individuals were paired in 7 combinations based on body size differences (either different sizes: >0.3 cm or equal sizes: <0.1 cm) and personality types (bold-aggressive and shy-submissive). The behaviors of all individuals were observed over a 10 min period during the pairing experiments. A successful pairing was defined as one where no aggressive behaviors resulted in fin damage, physical injury, or death during the 10 min observation. Any instance failing to meet these criteria was considered a pairing failure. 

To ensure that the pairing success rates were not affected by individuals’ previous experience within a short time, the experimental interval between two pairings on each fish was not less than 2 days. The pairing success rate of the combinations and each individual’s pairing result were analyzed and compared accordingly. Among successful pairings, the individual showing a higher frequency of appeasing and submissive behaviors was considered the loser, while the other individual was considered the winner. A total of 67 pairs were tested for their success rate, 44 of which had their behavior video recorded using a Canon camera (EOS 70D).

### 2.5. Statistical Analysis

In the behavioral ethogram recording experiments, the individual behaviors of *A. clarkii* were observed, summarized and categorized. The behavioral responses of *A. clarkii* were determined by the direct manual observation of the video. Firstly, various behaviors were identified and acquainted through repeated observation. Subsequently, two persons observed the videos of the personality test and the pairwise experiment independently to record the behavioral variables. Finally, the records of the two observers were compared, and any inconsistent results were subsequently re-examined to ensure the reliability of the findings.

The frequency of various behaviors in individuals were compared among the three groups by the Kruskal–Wallis test, as the variances of the observations were found to be heterogeneous. Additionally, comparisons were made between large and small individuals within the shared group using Student’s *t*-test. 

In the personality assessment experiments, the behaviors of *A. clarkii* individuals were recorded in two tests. Four behavioral variables were utilized for the personality assessment: the latency time of first attack on models, the frequency of biting on models, the latency time of the first attack on mirrors, and the frequency of attacking on mirrors. To normalize the data, the latency times to first attack (on models and mirrors) were log transformation and the frequencies of attacks or biting were square root transformed. Principal component analysis (PCA) was employed to condense the 4 recorded behaviors to fewer variables for assessing individual personality. Subsequently, individuals were classified into personality types using hierarchical clustering methods based on the personality assessment. The reliability of the personality assessment was tested by establishing the Fisher’s canonical discrimination formula to discriminate the personality type of each individual. The differences in individual behaviors between two personality type groups were examined using *t*-tests. Each individual underwent two rounds of personality assessments and consistency testing.

In the pairing experiments, differences in the pairing success rates among the 7 combinations were analyzed using Fisher’s exact test. Individual behaviors were then analyzed for their association with personality and body size using categorical principal components analysis (CATPCA), with the individual outcome (loser or winner) serving as a supplementary variable. Analysis of variance (ANOVA) was employed to compare the frequencies of individual behaviors (square root transformation), with body type and personality type considered as fixed factors. Student’s *t*-test was used to compare the behavioral frequencies between winners and losers in the fighting outcomes. Furthermore, the frequencies of behaviors within a pair were compared using Student’s *t*-test between successful and failed pairs, as well as between pairs with an equal body size and those without equal body sizes.

All statistical analysis was performed using IBM SPSS Statistics software version 25.

## 3. Results

### 3.1. Behavioural Ethogram

The individual behaviors of *A. clarkii* were observed and recorded within the three groups and summarized into 15 types (Table 1). Based on the possible intention behind each behavior, the behavioral ethogram was classified into four categories: (1)Normal behaviors: This category includes common behaviors such as foraging, shelter occupation, leaving the shelter, scouting, bite, and darting.(2)Threatening behaviors: This category encompasses behaviors such as dorsal leaning, lateral displaying, lunging, side-by-side swimming, and jaw clicking.(3)Appeasing behaviors: These behaviors, indicative of submission and pacifying opponent, include low-frequency trembling and high-frequency trembling.(4)Agonistic behaviors with aggressive intention: This category consists of behaviors like attack biting and chasing.

We focused on three categories related to social interaction: threatening, agonistic, and appeasing behaviors. The frequency of the occurrence of each of the seven typical behaviors (attack biting, chasing, lunging, side-by-side swimming, jaw clicking, low-frequency trembling and high-frequency trembling) were manually recorded and subsequently utilized for statistical analysis in follow-up experiments. Two behaviors, dorsal leaning and lateral displaying, which have been categorized as threatening behaviors, were not included in the analysis because of their lesser threatening intent. 

Individuals in the solitary group exhibited minimal signs of threatening, agonistic, or appeasing behaviors during the 10 min observations. Conversely, the frequency of both threatening and agonistic behaviors was significantly higher in the separated group compared to the shared group (*p* < 0.05) (Figure 1A). Moreover, significant differences were observed in the frequency of behaviors among individuals of different body sizes within the shared group: larger individuals were more prone to displaying agonistic and threatening behaviors, while smaller individuals predominantly exhibited appeasing behaviors (*p* < 0.05, Figure 1B).

### 3.2. Personality Assessment 

By testing the reaction to intruders, we found that *A. clarkii* individuals mainly exhibited two distinct personality types: bold–aggressive and shy-submissive. Individuals categorized as bold-aggressive displayed a high frequency of attack behaviors towards novel objects and intruders, demonstrating a proactive response to risks. In contrast, individuals categorized as shy-submissive tended to hide and showed a relatively high frequency of appeasing behaviors. Additionally, some individuals did not clearly exhibit either of these two distinct personality traits. 

Based on tests of their behavioral responses to models and mirrors, the Principal Component Analysis (PCA) of the four behavioral variables clearly depicted the behavioral traits of each individual. According to the cluster analysis, 41 individuals were categorized into two personality types (23 with bold-aggressive type personality and 18 with shy–submissive type personality), following the removal of seven individuals with inconsistent personality traits (see Figure 2 and clustering plot in Appendix A). Additionally, their personalities were successfully discriminated using Fisher’s canonical method (23 bold-aggressive and 18 shy-submissive), with significant behavioral differences (Appendix A). Notably, these 41 individuals exhibited personality consistency in the two personality testing experiments, while the seven individuals with inconsistent traits were excluded from subsequent pairing experiments. 

### 3.3. Pairing Result

The pairing success rates varied across different personality combinations (Figure 3). Interestingly, the combination of large and shy–submissive individuals paired with small and shy-submissive individuals (Ss) exhibited the highest success rate at 70%. Conversely, the combination of large and shy–submissive with small and bold-aggressive (Sb) showed the lowest success rate, at only 10% in the experiment.

In the pairing experiments, individual behaviors were found to be associated with body size and personality on different levels (Figure 4a). Notably, threatening and agonistic behaviors exhibited a significant positive correlation (Pearson’s r = 0.778, *p* < 0.01), and both were negatively influenced by a small body size, with no significant differences between the two personality types (Figure 4b). Moreover, the frequencies of appeasing behaviors were significantly higher in shy–submissive individuals compared to bold–aggressive individuals, and higher in small individuals compared to large and equal-sized ones (Figure 4b). In addition, the outcome of individuals in the fighting (winner or loser) after the paired experiment was included as a supplementary variable (Figure 4a), revealing a significantly higher frequency of appeasing behaviors in winners (*t*-test, *p* < 0.05). 

Pairing success was found to be associated with various behaviors exhibited by the two interacting individuals within a pair. Remarkably, successful pairs exhibited a significantly higher total frequency of appeasing behaviors compared to failed pairs, while displaying a lower total frequency of agonistic behaviors than failed pairs (Figure 5a). Furthermore, there was a significant positive correlation between the rate of successful pairing and the total frequency of appeasing behaviors (Pearson r = 0.770, *p* = 0.043). Pairs of equal body size (eB, eS, and eBS) exhibited more frequent agonistic and threatening behaviors compared to pairs of unequal body size (Ss, Bs, Bb, and Sb) (Figure 5b). Additionally, there was a highly significant positive correlation between the total frequency of threatening behaviors and the total frequency of agonistic behaviors within a pair (r = 0.929, *p* = 0.003).

## 4. Discussion

The research framework proposed by Tinbergen [27], which emphasizes the importance of observing and describing organisms, remains the cornerstone methodology for guiding and organizing behavioral research. In line with this framework, our study recorded the behavior ethogram of *A. clarkii* individuals under laboratory conditions. Building upon this behavior ethogram, we analyzed their social interaction activities, including agonistic, threatening, and appeasing behaviors. Nowadays, many behavioral studies usually use software to automatically identify and analyze the behavioral characteristics of animals, avoiding the bias that can be caused by artificial judgement. In our study, although traditional observation by eyes was still used, to avoid subjectivity, we had two persons observe the animals separately and then check and confirm the results to ensure the reliability of the results.

### 4.1. Interactive Behaviours in A. clarkii

In our study, we summarized the behavioral ethogram of *A. clarkii* at an early growth stage under laboratory conditions. Notably, we observed types of behavior similar to those reported by Moyer and Bell [24] in mature individuals, such as jaw clicking and attack biting. We found that these different behaviors had distinct meanings. For instance, side-by-side swimming, identified as a threatening behavior, is interpreted as a means through which individuals exaggerate their body size and assess each other’s strength, usually preceding threats or attacks [28]. Conversely, attack bites pose a significant risk of causing injury or even death to the fish. Similarly, jaw clicking is considered an aggressive vocal behavior of *A. clarkii* [29], and research by Colleye et al. [30] has demonstrated that this vocal behavior can convey information about an individual’s body size through the volume of the sound produced. Such communication is of substantial importance for members of social groups engaged in competition for and the maintenance of social status [30,31,32].

We observed variations in the frequency of different types of behaviors across different social contexts (Figure 1). In solitary groups, where there was no social interaction among individuals, the frequency of threatening, agonistic, and appeasing behaviors remained relatively low. However, in shared groups, stable hierarchical relationships were quickly established within the first 2 min due to direct face-to-face contact between individuals, so consequently, over the next 10 min, both large (usually dominant one) and small (usually subordinate one) individuals exhibited lower frequencies of threatening, agonistic, and appeasing behaviors (Figure 1B). These behaviors may indicate that larger individuals frequently exhibited threatening and attacking signals to maintain their social status, whereas smaller individuals, at a disadvantage in terms of body size, displayed obedience and appeasing behaviors to avoid potential attacks. In separated groups, where individuals were isolated and could only communicate visually through glass, the frequencies of threatening and agonistic behaviors were significantly higher compared to the shared group. This increased frequency can be attributed to the individuals being in a state of competition for hierarchical status, as they were less likely to quickly establish hierarchical relationships through visual communication alone. The most frequently observed threatening behaviors included lunging and side-by-side swimming, conveying a stronger threatening intent. Additionally, the frequency of attacking bites through the glass was extremely high in the agonistic behaviors.

### 4.2. Personality of A. clarkii

Personality differences have been documented among individual anemonefish [6], and similar behavioral variations between individuals were observed in our study. While some studies suggest that animal personality can be represented as a continuous variable, such as shyness–boldness [7], certain behavioral traits tend to be correlated among individuals [7,8,33]. For instance, the boldness of *Thymallus thymallus* has been found to be positively correlated with aggressiveness [10]. 

Thus, in our study, we utilized principal component analysis (PCA) to reduce the four behavioral variables of the clownfish subjects. Subsequently, we categorized individuals (41 out of 48) with distinct behavioral traits into two personality types: the bold–aggressive type (characterized by greater activity and frequent attacks on novelty and intruders) and the shy–submissive type (exhibiting high-frequency appeasement and fewer or no attacks on novelty and intruders). This simplification of personality types into two categories facilitated the subsequent study of the effects of personality type and body size on competition for social hierarchies in pairing experiments. 

### 4.3. Behaviours in the Process of Hierarchical Establishment

Anemonefish typically adhere to a strict social hierarchy, wherein individuals are commonly ranked based on their body size [34,35]. In our study, large individuals predominantly displayed lunging behavior, a typical threatening behavior but with less threatening intent, similar to the observations of Iwata and Manbo [28]. Conversely, small ones mainly resorted to appeasing behavior, including low-frequency trembling. This is because individuals would avoid costly aggressive behaviors by assessing their competitors using nonaggressive means [36,37]. For small individuals, the loss they face may not only be a lowered rank, but may even include unnecessary energy depletion, physical injures, etc. Therefore, small individuals are very likely to display a low frequency of threatening behaviors and a high frequency of appeasing behaviors towards large individuals [38]. In successful pairs, larger individuals consistently emerge as the winners, exhibiting more agonistic and threatening behaviors and fewer appeasing behaviors (Figure 4), underscoring the significant impact of body size on competition for social hierarchies [39]. Larger individuals possess an advantage in vying for hierarchical positions, yet it remains uncertain whether they can successfully establish stable hierarchical relationships with smaller individuals. 

We observed that in pairs where individuals differed in body size, the pairing success was significantly influenced by the personalities of the opponents, such as in pairs Ss vs. Sb and Bs vs. Sb (Figure 3). Shy-submissive individuals tended to exhibit more appeasing behaviors during pairing compared to bold-aggressive individuals (Figure 4b). However, some small and bold-aggressive individuals were disinclined to display appeasement and even continued to exhibit agonistic behaviors. This indicates that other than the importance of body size, the personality of anemonefish also plays a role in individual behaviors in competing for hierarchical position.

However, in pairs where individuals were of equal body size (eB, eS, and eBS), both opponents were unable to swiftly establish their social status based on body size alone. Consequently, they were more likely to compete for their social status by fighting, resulting in more frequent agonistic and threatening behaviours (Figure 5b). This phenomenon is reminiscent of observations in another clownfish *A. percula*, where conflict intensity was heightened when individuals were more similar in size [40]. Additionally, further support for this was evident in our aforementioned behavioural ethogram experiment, where greater levels of threatening and aggressive behaviour were observed in separated group (Figure 1B), as hierarchical status could not be immediately established. Under these circumstances, individual personality exerted a less significant effect on changes in success rates (Figure 3).

### 4.4. The Decisive Behaviour for the Establishment of Hierarchies

In the pairing experiment, the interactive behavior between immature individuals was primarily motivated by competition for social hierarchical status. A stable hierarchical relationship was likely to be established only if both individuals displayed more appeasing behaviors, as indicated by a significant positive correlation between the appeasing frequency and pairing success rate (Pearson r = 0.770). The pairing results further demonstrated that successful pairs were associated with the total frequency of agonistic and appeasing behaviors in both individuals, with more appeasing and less agonistic behaviors in successful pairs compared to failed pairs (Figure 5). For smaller individuals, being at a disadvantage in combat, they often exhibited high-frequency trembling (appeasing behaviors) towards larger individuals. In response, larger individuals typically refrained from threats and attacks against the small ones. Thus, frequent appeasing behaviors signaled the establishment of a hierarchical status for the larger individuals, with the small individuals no longer perceived as a threat. This stability in social relations led to successful pairing. Additionally, some dominant individuals were observed to exhibit appeasing behaviors to pacify the subordinate after the hierarchical relationship was established. Avoiding conflict in stable social relationships is certainly beneficial, as it avoids costly aggressive behavior, the risk of injuries and energy expenditure [16,41]. In contrast, when neither side compromised and could not be driven out due to the experimental setting, the stronger side was more inclined to attack or even kill the other. Pairing failure was typically attributed to either an insufficient frequency of appeasing behaviors or both individuals failing to decrease their agonistic behaviors, leaving their hierarchical status uncertain even after prolonged conflict. The results of this study also suggested that due to the differences in personality among anemonefish individuals, a reasonable combination of body size and personality should be taken into account during pair breeding to avoid losses caused by inter-individual fights and injuries [41].

Hattori [42] observed frequent migrations and invasions between host anemones of immature *A. clarkii* individuals in Okinawa Island waters (field study). Such migrations and invasions were even more common in sea areas with high anemone densities [19,43]. These observations suggest that immature individuals often engage in fights to occupy a higher social hierarchical status and access more resources. Based on the personality type and behaviors of *A. clarkii*, it can be speculated that in natural sea areas, when immature small individuals migrate among host anemones, shy–submissive individuals are more likely to be accepted and avoid the risk of being attacked and bitten due to their appeasing behaviors towards larger individuals, thus benefiting individual survival; meanwhile, bold–aggressive individuals are more likely to be driven away in search of another host anemone due to their unwillingness to settle for a lower-ranked status, thereby contributing to population dispersal and enhancing the chances of gene exchange between populations. Therefore, both bold–aggressive and shy–submissive personalities contribute to the population survival of anemonefishes.

## 5. Conclusions

Our results suggested there are two distinct personality types in *Amphiprion clarkii:* bold–aggressive and shy–submissive. The impact of personality on the establishment of a stable social hierarchy was confirmed, with the frequency of appeasing behaviors being the main factor influencing the success rate. These personality-related behavioral traits are potentially important for both individual and population survival, and extend our understanding of the establishment and evolution of social hierarchies in anemone fishes and their impact on population dispersal. Furthermore, they have the potential ability to guide aquaculture practices.

## Figures and Tables

**Figure 1 animals-14-02216-f001:**
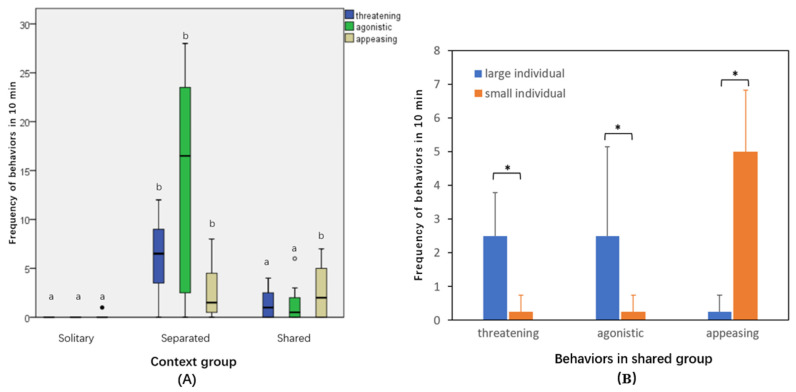
Frequency of behaviors in *Amphiprion clarkii*: (**A**) boxplots in three context groups: solitary group (*n* = 8), separated group (*n* = 12), and shared group (*n* = 8). Different letters on the top of boxplots indicate significant differences among groups (*p* < 0.05). (**B**) mean frequencies in large and small individuals in shared group, divided by body size. Error bars indicate standard deviations (*n* = 4). Asterisk (*) on the top of bars indicate significantly differences between two body size subgroups (*p* < 0.05).

**Figure 2 animals-14-02216-f002:**
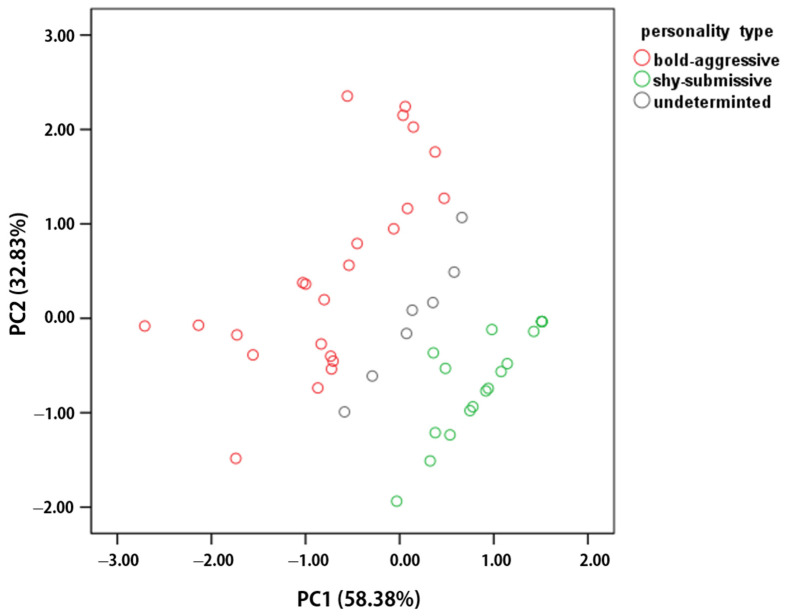
Biplot of principal component analysis (PCA) and clustering results of personality assessment of *Amphiprion clarkii*. Based on the results of the cluster analysis, the individuals could be mainly categorized into a bold–aggressive personality (red circle, *n* = 23) and a shy–submissive personality (green circle, *n* = 18), with the rest of undetermined individuals having no distinct personality traits (gray circle, *n* = 7).

**Figure 3 animals-14-02216-f003:**
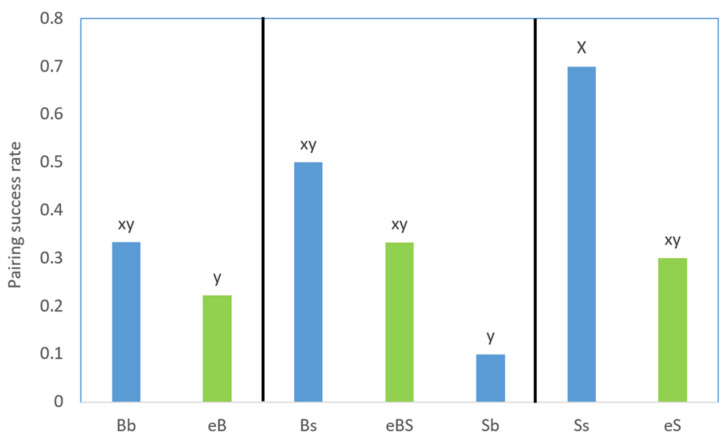
The pairing success rates in personality and size combinations. Blue bars represent pairs of different sizes, green ones of equal sizes. (**Left**): two bold–aggressive; (**Middle**): one bold-aggressive and one shy–submissive; (**Right**): two shy-submissive. Notes for pairing combination symbols: B or b, bold-aggressive personality with large or small body size in a pair; S or s, shy–submissive personality with large or small body size in a pair; e, equal body sizes in a pair. Different letters above bars indicate a significant difference in the pairing success rate between combinations (Fisher’s exact test, *p* < 0.05), and the same letter indicates that the difference was not significant at *p* > 0.05).

**Figure 4 animals-14-02216-f004:**
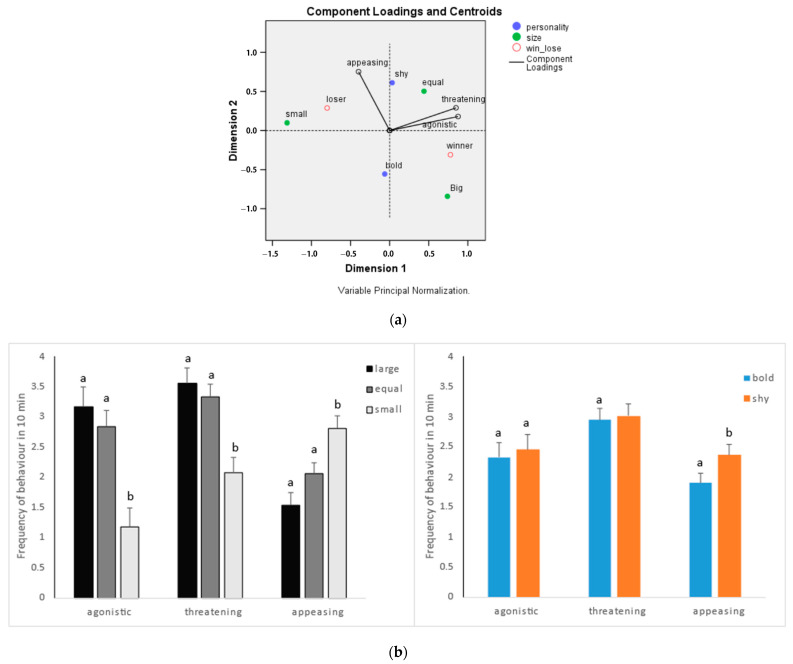
Individual behavior in relation to body size and personality in pairing experiment. (**a**) Categorical principal components analysis diagram. The total variance accounted for across the two components is 61.8% (indicating reasonable fit), with total Eigenvalue = 3.089. (**b**) Estimated marginal means of behavioral frequencies of individuals. (**Left**): individual behavior with different body size in a pair; (**Right**): individual behavior with different personality. Values were square root transformed, and the lines above bars were standard errors. Different letters above bars correspond to significant differences (*p* < 0.05).

**Figure 5 animals-14-02216-f005:**
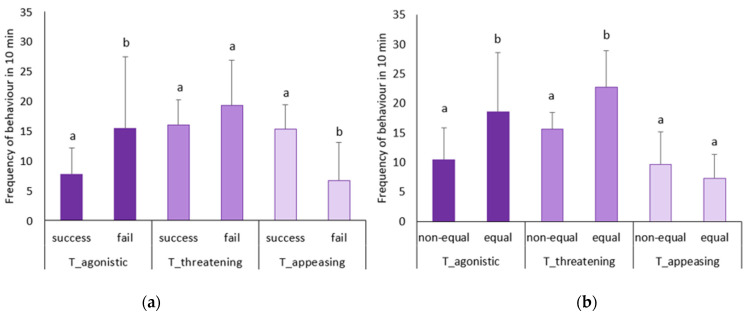
Frequency of various behaviors (means ± SD) in failed and successful pairs (**a**) and in equal body size and non-equal body size pairs (**b**). Failed pair, *n* = 34; success pair, *n* = 10; equal body size, *n* = 18; nonequal body size, *n* = 26. Different letters above bars indicate significant difference (*p* < 0.05).

**Table 1 animals-14-02216-t001:** Behavioral ethogram of *Amphiprion clarkii*.

Behaviour	Description	Category
Shelter occupation	Staying in the PVC tube for a long time.	normal behaviors [24]
Leaving the shelter	Leaving the PVC tube and swimming in the environment.	normal behaviors [15]
Scouting	Swimming around the PVC tube and returning.	normal behaviors [15]
Foraging	Opening the mouth and swallowing food.	normal behaviors [15,24]
Bite	Slight biting another individual without physical damage.	normal behaviors [24]
Darting	Swimming suddenly without purpose and direction, neither to avoid chasing nor back toward shelter.	normal behaviors [15,24]
Dorsal leaning	Dorsal fin is upright and unfolded with a less threatening intention	threatening behaviors [24]
Lateral display	Facing another individual, turning the body to the side and maintaining a stiff position, sometimes accompanied by an upright fin, with a less threatening intention.	threatening behaviors [24]
Lunging *	Rushing to another individual with a threatening intention.	threatening behaviors [24]
Side-by-side swimming *	Two individuals swimming side by side with a moderate threatening intention.	threatening behaviors [24]
Jaw clicking *	Clicking the upper and lower jaw to make the head tremble with a strong threatening intention.	threatening behaviors [24]
Attack biting *	Gnawing at another individual violently, causing injury.	agonistic behaviors [15,24]
Chasing *	Following another individual, may be accompanied by attacking and biting with a strong aggressive intention.	agonistic behaviors [15,24]
Low-frequency trembling *	Starting from the head, the trembles quickly spreading throughout the body with a frequency of 1–2 times per second with a moderate appeasing intention.	appeasing behaviors [24]
High-frequency trembling *	Starting from the head, the trembles quickly spreading throughout the body with a frequency of 4–6 times per second with a strong appeasing intention.	appeasing behaviors [24]

* The frequencies of these behaviours were manually recorded during the observations. The number in brackets means the same description was used in this reference.

## Data Availability

All data included in this study are available upon request from the corresponding author.

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
