# Peer review of "Effects of Personality and Behavioral Syndromes on Competition for Social Hierarchical Status in Anemonefish Amphiprion clarkii"

_animals, 2024, doi:10.3390/ani14152216_

Round 1

Reviewer 1 Report

Comments and Suggestions for Authors

The study extends our understanding of the establishment and evolution of social hierarchies in anemone fishes and their  impact on population dispersal.  Furthermore,  they have potential applications to guide aquaculture practices. The article is concise and logical and can be accepted  after completing the following minor changes:

“the latency time of first attack on novel objects, the frequency of attacks on novel objects,   the latency time of the first attack on intruders,  and the frequency of biting intruders.” Please give the definition and calculation method of these parameters.

In addition, are behavioral parameters recorded manually?  Or software?  The method section needs to be supplemented.

Author Response

Comments 1: “the latency time of first attack on novel objects, the frequency of attacks on novel objects,   the latency time of the first attack on intruders,  and the frequency of biting intruders.” Please give the definition and calculation method of these parameters.

In addition, are behavioral parameters recorded manually?  Or software?  The method section needs to be supplemented.

Response 1: Thank you for pointing this out. All behavioral parameters were recorded manually. We have revised it.

Line 133  --" Behavioural recordings of individuals in these 3 groups were manually observed, described and summarized in a behavioural ethogram. "

Line 137-142  --“two tests were used to assess the personality of each A. clarkii individual, specifically their reaction to two conspecific intruders. The first was an anemonefish model (of the same size as the examined individuals) introduced into the tank, and the second was a mirror image of itself (a mirror was placed in the tank). The latency time to first attack and the frequency of attacks on the model and mirror were observed and recorded within a 10-min period.”

Line 171-177  --“Behavioural responses of A. clarkii were determined by direct manual observation of video. Firstly, various behaviours were identified and acquainted through repeated observation. Subsequently, two persons observed the videos of the personality test and the pairwise experiment independently to record the behavioural variables. Finally, the records of the two observers were compared, and any inconsistent results were subsequently re-examined to ensure the reliability of the findings..”   

Reviewer 2 Report

Comments and Suggestions for Authors

The manuscript was well designed and conducted. It is also well written. I think it almost can be accepted for publish at the present stage. I only have a few minor comments.

Line 227-223, There are 15 types of behavior that can be classified into four categories. The authors selected seven out of 15 behavior types, all belong to three categories. All types of normal behavior and two of threatening behavior are excluded. It is generally well written. However, it would be nice if the authors can re-organize these sentences a little bit.

Figure 1. what is the unit of y-axis. Times per min? or times per 10 min (i.e. total times of activities during 10 min observation)?

Line 282 ‘small body size’, delete ‘small’

Figure 4, 5, there are some format issues. The figures need to be redrawn. Panel a, b, c might be more proper for figure 4. Important information such as sample size, what is the bar means, what are those letters mean et al. are missing.

Author Response

We would like to express our sincere gratitude to you for your hard work and professional comments in reviewing our manuscript.

Comments 1: Line 227-223, There are 15 types of behavior that can be classified into four categories. The authors selected seven out of 15 behavior types, all belong to three categories. All types of normal behavior and two of threatening behavior are excluded. It is generally well written. However, it would be nice if the authors can re-organize these sentences a little bit.

Response 1: Thank you for pointing this out.  We have revised it according to your coment. Line 221-225 --  “We focused on three categories with social interaction intention: threatening, agonistic, and appeasing behaviours. The frequency of occurrences for each of seven typical behaviours (attack biting, chasing, lunging, side-by-side swimming, jaw clicking, low-frequency trembling and high-frequency trembling) in three categories were recorded manually and subsequently utilized for statistical analysis in follow-up experiments.”

Comments 2: Figure 1. what is the unit of y-axis. Times per min? or times per 10 min (i.e. total times of activities during 10 min observation)?

Response 2: Thank you for pointing this out. The unit of y-axis is total frequencies of behaviours during 10 min observation.

Line 232-233 -- “Individuals in the solitary group exhibited minimal signs of threatening, agonistic, or appeasing behaviours during 10 min observations.”

Comments 3: Line 282 ‘small body size’, delete ‘small’

Response 3: Thank you for pointing this out. We didn't make ourselves clear, so we have revised it.

Line 288-289 --“ and both were negatively influenced by small body size, with no significant differences between the two personality types.”

Comments 4: Figure 4, 5, there are some format issues. The figures need to be redrawn. Panel a, b, c might be more proper for figure 4. Important information such as sample size, what is the bar means, what are those letters mean et al. are missing.

Response 4: Thank you for pointing this out. We have redrawn Figure 4 and 5 according to your comments.

Reviewer 3 Report

Comments and Suggestions for Authors

Animals review 30.06.24

Effects of personality and behavioural syndromes on competition for social hierarchical status in anemonefish Amphiprion clarkii

Wu et al.

The authors carried out a study to describe personality and test the effect of individual differences in behaviour on the establishment of stable social hierarchies. They found that individuals with shy personality types are more likely to be successful in establishing a stable hierarchy with another individual. Body size also matters; relatively larger shy individuals matched with smaller bold individuals has the lowest chance of successful pairing.

I am not convinced that a novel object test using a model of a conspecific can really be called a novel object test, especially as the behaviours measured were not different from those in the mirror test for agnostic behaviour. The two experiments appear to be different versions of the simulated intruder test. For a novel object test the object should be completely novel, and is not intended to elicit aggressive or social interaction, rather used to measures boldness/exploratory tendency in a non-social context.

Having said this, I think the results from both experiments can still be used and it doesn´t change the overall message in terms of aggressive personality and pairing success, but I suggest a change of wording.

The study is interesting because it provides evidence of a mechanism whereby different personality types may have different dispersal rates from a population, which can impact a range of evolutionary processes.

General comments

Figures and figure legends could be presented more clearly (including supplementary).

Specific comments

26-27: each individual

75: increased more food

96: other vertebrates

97-100: these lines seem a bit out of place here, could be incorporated into methods section

For clarity you could state your hypotheses at the end of the intro (after reading literature, what specifically do you expect from your investigation on influence of personality and body size on the success of establishing social hierarchies)

108-119: any idea how many generations they have been bred in captivity, and whether all individuals are closely related or not?

135-136: adaptation and adjustment of behaviors = acclimation

140: were properly used

141-144: is test 1 really a novel object test? Test 1 & 2 both test the latency and rate of attacks to a size matched intruder (whether it be a model or mirror image). Novel object tests to look at exploratory tendency and/or boldness do not usually use models of conspecifics, which are likely to induce other (sociability / aggressiveness) behaviours.

171: conducted tested

180-182: I would argue that these are two behaviours (see comment for lines 141-144)

184: log transformed

199: utilised used

235: Figure 1. (A)

236: alphabets letters

236: significantly differences among each groups

242: By testing the reaction to novel objects and intruders we found that…

252: significant?

267: different personality combinations

270: nearly? What was the actual lowest success combination?

273: This figure is not very easy to interpret, can you display the results differently? or give a clearer explanation in the legend?

287-288: significantly higher frequency of appeasing behaviours in winners

308: non-equal

309 → discussion need some work. Some of the text here could be better used in the intro or methods

Hope you find this review helpful

Comments on the Quality of English Language

Please be consistent with use of British or American spelling. There is currently a random mix of behaviour/behavioural and behavior/behavioral, for example.

The text in places could also be more succinct.

Author Response

We would like to express our sincere gratitude to you for your hard work and professional comments in reviewing our manuscript.

Comments 1: I am not convinced that a novel object test using a model of a conspecific can really be called a novel object test, especially as the behaviours measured were not different from those in the mirror test for agnostic behaviour. The two experiments appear to be different versions of the simulated intruder test. For a novel object test the object should be completely novel, and is not intended to elicit aggressive or social interaction, rather used to measures boldness/exploratory tendency in a non-social context.

Having said this, I think the results from both experiments can still be used and it doesn´t change the overall message in terms of aggressive personality and pairing success, but I suggest a change of wording.

The study is interesting because it provides evidence of a mechanism whereby different personality types may have different dispersal rates from a population, which can impact a range of evolutionary processes.

Response 1: We totally agree with your comment. Yes. The two experiments appear to be different versions of the simulated intruder test. We have revised it.

Line 141-146 – “two tests were used to assess the personality of each A. clarkii individual, specifically their reaction to two conspecific intruders. The first was an anemonefish model (of the same size as the examined individuals) introduced into the tank, and the second was a mirror image of itself (a mirror was placed in the tank). The latency time to first attack and the frequency of attacks on the model and mirror were observed and recorded within a 10-min period.”

Comments 2: General comments

Figures and figure legends could be presented more clearly (including supplementary).

Response 2:

Thank you for pointing them out. we revised Figure 3, Figure 4, Figure 5 and Figure S2

Comments 3:

Specific comments

26-27: each individual

75: increased more food

96: other vertebrates

Response 3:

Thank you for pointing them out. we revised them as your comments.

Comments 4:

97-100: these lines seem a bit out of place here, could be incorporated into methods section

Response 4:

Thank you for pointing them out. We have rearranged text in Lines 85-102.

Comments 5: For clarity you could state your hypotheses at the end of the intro (after reading literature, what specifically do you expect from your investigation on influence of personality and body size on the success of establishing social hierarchies)

Response 5:

Thanks to your comments. We have made some modifications in introduction.

Comments 6:

108-119: any idea how many generations they have been bred in captivity, and whether all individuals are closely related or not?

Response 6:

All A. clarkii used for this investigation were F1 descendants of wild-caught, and they are not closely related.

Comments 7:

135-136: adaptation and adjustment of behaviors = acclimation

140: were properly used

Response 7:

Thank you for pointing them out. we revised them as your comments.

Comments 8:

141-144: is test 1 really a novel object test? Test 1 & 2 both test the latency and rate of attacks to a size matched intruder (whether it be a model or mirror image). Novel object tests to look at exploratory tendency and/or boldness do not usually use models of conspecifics, which are likely to induce other (sociability / aggressiveness) behaviours.

Response 8: We totally agree with your comment. Yes. The two experiments appear to be different versions of the simulated intruder test. We have revised it.

Line 141-146 – “two tests were used to assess the personality of each A. clarkii individual, specifically their reaction to two conspecific intruders. The first was an anemonefish model (of the same size as the examined individuals) introduced into the tank, and the second was a mirror image of itself (a mirror was placed in the tank). The latency time to first attack and the frequency of attacks on the model and mirror were observed and recorded within a 10-min period.”

Comments 9:

171: conducted tested

Response 9:

Thank you for pointing them out. we revised them as your comments.

Comments 10:

180-182: I would argue that these are two behaviours (see comment for lines 141-144)

Response 10:

We totally agree with your comment. We have revised in Line 141-146.

Comments 11:

184: log transformed

199: utilised used

Response 11:

Thank you for pointing them out. we revised them as your comments.

Comments 12:

235: Figure 1. (A)

236: alphabets letters

236: significantly differences among each groups

242: By testing the reaction to novel objects and intruders we found that…

Response 12:

Thank you for pointing them out. we revised them as your comments.

Comments 13:

252: significant?

Response 13:

Thank you for pointing it out. We have rewrite it in line 257 – “41 individuals were categorized into two personality types (23 with bold-aggressive type personality and 18 with shy-submissive type personality),” and in line 260 –“with significant behavioural differences (Figure S2).”

Comments 14:

267: different personality combinations

Response 14:

Thank you for pointing it out. We have revised it.

Comments 15:

270: nearly? What was the actual lowest success combination?

Response 15:

Thank you for pointing it out. The word “nearly” should be deleted. And the actual lowest success combination was  only 10%.

Comments 16:

273: This figure is not very easy to interpret, can you display the results differently? or give a clearer explanation in the legend?

Response 16:

Thank you for pointing it out. We have redrawn  Figure 3.

Comments 17:

287-288: significantly higher frequency of appeasing behaviours in winners

308: non-equal

Response 17:

Thank you for pointing it out. We have revised them according to your comments.

Comments 18:

309 → discussion need some work. Some of the text here could be better used in the intro or methods

Response 18:

Thank you for your comment. We have revised in Line 330-338.

“In line with this framework, our study recorded the behaviour ethogram of A. clarkii individuals under laboratory conditions. Building upon this behaviour ethogram, we analyzed their social interaction activities, including agonistic, threatening, and appeasing behaviours. Nowadays, many behavioural studies usually use software to automatically identify and analyse the behavioural characteristics of animals, avoiding the bias that can be caused by artificial judgement. In our study, although traditional observation by eyes was still used, to avoid subjectivity, we had two persons observe the animals separately and then check and confirm the results to ensure the reliability of the results.”

Comments 19:

 Comments on the Quality of English Language

Please be consistent with use of British or American spelling. There is currently a random mix of behaviour/behavioural and behavior/behavioral, for example.

The text in places could also be more succinct.

Response 19:

Thank you for pointing it out. We have revised them.

Reviewer 4 Report

Comments and Suggestions for Authors

This paper is about the effect of behavioural syndromes on the success of establishing social hierarchies in anemonefish. The authors also summarized the behavioral ethogram of Amphiprion clarkii. The authors have clearly spent a lot of time in conducting this study and show interesting results. I think the manuscript would benefit from a thorough rewriting, bringing the new insights forward more clearly.

Comments are mentioned bellow accordingly:

Abstract, title and references:

The title is informative and relevant based on the content of the manuscript. The aim of research is clear and in accordance with the tile of the study; on the page 2 lines: 85-86 mentioned " The aim of the present study is to investigate whether anemonefish personality and behaviours influence the success of establishing social hierarchies."What the study found and how they did it is not clear yet. The references are not recent (only 5 out of 40 references are cited 2020 afterwards) and key studies are not included in the reference list.

on the page 1, line 33: " ... it can be speculated..." this kind of sentences should be mentioned in the discussion part not in the abstract.

On the page 1, line 32 "Appeasing behaviour" there is no clear and explicit definition and justification in the manuscript and therefore should be removed from the abstract accordingly. Do fish show appeasing behaviours? if so you need to come with some new and updated references.

Introduction/background:

It seems the uploaded file of the manuscript is not the final version for review as there are already few comments in.

It is clear what is already known about the topic.  Although, the research questions are not clearly outlined; " what is the potential significance of these personality-related behavioural traits 90
in anemonefish for both individual and population survival? " Do you think you can answer to this question based on your results and experimental observation? if so please justify and add the relevant results.

on the page 3, lines 97-106, the last paragraph of the introduction should be your aims and research questions. So you need to merge this into the earlier paragraph. 

Methods

The process of subject selections is clear and the variables defined and measured appropriately. Although the study methods validity and reliability is not clear and well explained. The authors should address this issue with more clarifications with valid references in methodology section.

Moreover there is no enough detail in the methodology section in order to replicate the study. The authors can provide more details about each parts and defining variables also drawing experimental set up schemes and adding experimental time frames accordingly.

0n page 3, line 109: the country name were the experiment been performed should be included.

on page 3, line 111: " with body lengths of 1.8-3.0 cm" please ad +-SD of mean for the body length of the used animals.

On page 3 lines:113-114: " Polyvinyl chloride 113
(PVC) tubes served as shelters and habitats in the experiments" any references to use this material for sheltering? does probably a PVC shelter leakage any chemical contamination in your arena? justification in methodology is highly recommended with valid references. please add them accordingly.

on page 3, line 115: "...A. clarkii individuals were fed artificial compound feed..." please give details of the food grad and company

On page 3, lines: 121- 123: " Recordings of A. clarkii behaviors were captured using a Canon camera (EOS 70D) in recording mode." How you analyzed /processed behavioural responses?by the experimenter eyes counting/ observation on video files  or you used any manual or authomatic softwares to track your fish? It should be clarified accordingly. A schematic view /time line of the experiments procedures also  schematic views of experimental set up can help readers to understand more quickly an overview of the study. Please add them accordingly in the revised version.

The personality tests need to be more clarified and explained. No entries or definition about the methods used for these tests.

on page 3, lines: 133-135: " As there were no significant differences observed in behaviors between morning and afternoon, behavioral recordings were conducted once daily." There are studies that show fish activities and behaviour rates are different in the morning and afternoon. Did you any analysis to explain this result which would be also interesting. Also are there other studies finding the dame patterns? You need to elaborate this issue in the discussion part accordingly.

on page 3, line 135: "... with the initial 2 minutes allocated for adaptation..."

please explain why you use "adaptation" term? Why you do not use "acclimation" instead? The authors need to consider relevant and appropriate terminology in their methodological approaches.

on  the page 3, line: 142 A picture or model of a companion fish is a conspecific, not a novel object so there is a methodological issue in this part of your study. Otherwise if you have any valid references to justify this issue. If not then you need to edit and revise it as using a conspecific model object.

Results:

in general the data were presented in an appropriate way but table 1 and figure 4 need revisions as mentioned bellow:

On the page 5, line: 225: for the table showing behavioural ethograms please add any references that they also used the same description per each behaviour for the same or other species (also needs to be mentioned on the table) by adding another column on the table.

on the page 6, line: 247: " ...appeasing behavior..." there is no indication of definition for this behaviour in the methods section, please provide more info about it in the text or in the table 1.

on the page 6, lines: 247-248: ".... Additionally, some individuals did not clearly exhibit either of these two distinct personality traits. ..."  did you analyses further these individuals (other behavioural measurements/tests)?

on page 9, figure 4: please change figure number from "a" to "A" and the same for "b" as you referred them in the text accordingly.

Discussion and Conclusions:

there is no declaration of the limitations of the study. Also all the results need to be discussed from multiple angles and placed into context in separate paragraphs and /or well structures parts in the discussion section. More relevant references need to be included in the discussion section.

there is no conclusions included.

Comments on the Quality of English Language

Minor editing of English language required

Author Response

We would like to express our sincere gratitude to you for your hard work and professional comments in reviewing our manuscript.

Comment 1:

Abstract, title and references:

The title is informative and relevant based on the content of the manuscript. The aim of research is clear and in accordance with the tile of the study; on the page 2 lines: 85-86 mentioned " The aim of the present study is to investigate whether anemonefish personality and behaviours influence the success of establishing social hierarchies."What the study found and how they did it is not clear yet. The references are not recent (only 5 out of 40 references are cited 2020 afterwards) and key studies are not included in the reference list.

Response 1:

Thanks to your comment. We referenced recent literature on the subject.

Yllan, L., Heatwole, S., Wong, M., Rueger, T., 2024. Effect of social context on behaviour in anemonefish hierarchies. Animal Behaviour 209, 83–93. https://doi.org/10.1016/j.anbehav.2023.12.014

Kashimoto, R., Mercader, M., Zwahlen, J., Miura, S., Tanimoto, M., Yanagi, K., Reimer, J.D., Khalturin, K., Laudet, V., 2024. Anemonefish are better taxonomists than humans. Current Biology 34, R193–R194. https://doi.org/10.1016/j.cub.2023.07.051.

Hobson, E.A., Mønster, D., DeDeo, S., 2021. Aggression heuristics underlie animal dominance hierarchies and provide evidence of group-level social information. Proc. Natl. Acad. Sci. U.S.A. 118, e2022912118. https://doi.org/10.1073/pnas.2022912118.

Comment 2:

on the page 1, line 33: " ... it can be speculated..." this kind of sentences should be mentioned in the discussion part not in the abstract.

Response 2:

Thanks to your comment. We have revised it.

Our results suggested that in natural waters, when juvenile individuals migrate among host anemones, shy-submissive individuals are more likely to be accepted due to their appeasing behaviours towards larger individuals, thus avoiding the risk of being attacked and bitten, and benefiting the survival of the individual.

Comment 3:

On the page 1, line 32 "Appeasing behaviour" there is no clear and explicit definition and justification in the manuscript and therefore should be removed from the abstract accordingly. Do fish show appeasing behaviours? if so you need to come with some new and updated references.

Response 3:

Thanks to your comment. In contrast to the term "submissive," which is frequently employed by numerous researchers, we have selected the term "appeasing" to describe our chosen behaviour. This is because we consider "appeasing" to be a term that encompasses both the notion of submissiveness and the act of pacifying one's opponent.

Comment 4:

Introduction/background:

It seems the uploaded file of the manuscript is not the final version for review as there are already few comments in.

Response 4:

Thanks to your comment. The manuscript is the final version we submitted. The comments in the manuscript are the editor's cues to us.

Comment 5:

It is clear what is already known about the topic.  Although, the research questions are not clearly outlined; " what is the potential significance of these personality-related behavioural traits in anemonefish for both individual and population survival? " Do you think you can answer to this question based on your results and experimental observation? if so please justify and add the relevant results.

Response 5:

Thanks to your comment. Through our hierarchical establishment experiments (pairing success), we found that clownfish with different personalities behaved significantly differently in determining hierarchical status with other individuals, and that the personalities of both sides had a significant effect on the success of the pairing.

Comment 6:

on the page 3, lines 97-106, the last paragraph of the introduction should be your aims and research questions. So you need to merge this into the earlier paragraph. 

Response 6:

Thanks to your comment. We have rearranged the text in Line 85-101.

Comment 7:

Methods

The process of subject selections is clear and the variables defined and measured appropriately. Although the study methods validity and reliability is not clear and well explained. The authors should address this issue with more clarifications with valid references in methodology section.

Moreover there is no enough detail in the methodology section in order to replicate the study. The authors can provide more details about each parts and defining variables also drawing experimental set up schemes and adding experimental time frames accordingly.

Response 7:

Thanks to your comment. We have added more details in methodology section.

Comment 9:

0n page 3, line 109: the country name were the experiment been performed should be included.

Response 9:

Thanks to your comment. We have added it in Line 110 -- China.

Comment 10:

on page 3, line 111: " with body lengths of 1.8-3.0 cm" please ad +-SD of mean for the body length of the used animals.

Response 10:

Thanks to your comment. We have added it in Line 112 -- (2.4 ± 0.3 cm)

Comment 11:

On page 3 lines:113-114: " Polyvinyl chloride 113
(PVC) tubes served as shelters and habitats in the experiments" any references to use this material for sheltering? does probably a PVC shelter leakage any chemical contamination in your arena? justification in methodology is highly recommended with valid references. please add them accordingly.

Response 11:

Thanks to your comment. PVC pipe, made from polyvinyl chloride polymer, is commonly used in aquaculture and hydroponic systems due to its excellent chemical resistance. Although there is a risk of chemical release when exposed to sunlight or acidic water environments, all cultivation processes in this experiment were conducted indoors. The water quality was maintained at a stable pH of around 8.0, eliminating the risk of compound release. PVC pipes are not only widely used in the aquaculture industry but also in fish behavioral studies. For instance, as referenced in the following two studies, they are utilized in systems for long-term observation of self-feeding triggered activities in European seabass, and in the observation of fish behavior in mixed culture systems using artificial intelligence.

  1. Covès, et al., Long-term monitoring of individual fish triggering activity on a self-feeding system: An example using European sea bass (Dicentrarchus labrax), Aquacullture, 253(2006):385-392;

Jun Hu, et al., Real-time nondestructive fish behavior detecting in mixed polyculture system using deep-learning and low-cost devices, Expert Systems With Applications, 178(2021): 115051

Comment 12:

on page 3, line 115: "...A. clarkii individuals were fed artificial compound feed..." please give details of the food grad and company

Response 12:

Thanks to your comment. We have added it in Line 116-118 –“A. clarkii individuals were fed commercial pellets (45.5% crude protein, 10.5% lipid, 9.5% moisture and 9.8% ash, purchased from Xiamen Jiakang Feed Co., LTD, China) daily throughout the experiment,”

Comment 13:

On page 3, lines: 121- 123: " Recordings of A. clarkii behaviors were captured using a Canon camera (EOS 70D) in recording mode." How you analyzed /processed behavioural responses?by the experimenter eyes counting/ observation on video files  or you used any manual or authomatic softwares to track your fish? It should be clarified accordingly. A schematic view /time line of the experiments procedures also  schematic views of experimental set up can help readers to understand more quickly an overview of the study. Please add them accordingly in the revised version.

Response 13:

Thanks to your comment. We have added it in Line 175-181. – “Behavioural responses of A. clarkii were determined by direct manual observation of video (by eyes). Firstly, various behaviours were identified and acquainted through repeated observation. Subsequently, two persons observed the videos of the personality test and the pairwise experiment independently to record the behavioural variables. Finally, the records of the two observers were compared, and any inconsistent results were subsequently re-examined to ensure the reliability of the findings.”

Comment 14:

The personality tests need to be more clarified and explained. No entries or definition about the methods used for these tests.

Response 14:

Thanks to your comment. We have revised it.

 Line 136-142:

 “According to the methodology of personality traits testing used in previous studies [25,26], two tests were used to assess the personality of each A. clarkii individual, specifically their reaction to two conspecific intruders. The first was an anemonefish model (of the same size as the examined individuals) introduced into the tank, and the second was a mirror image of itself (a mirror was placed in the tank). The latency time to first attack and the frequency of attacks on the model and mirror were observed and recorded within a 10-min period.”

line 171-177:

“Behavioural responses of A. clarkii were determined by direct manual observation of video. Firstly, various behaviours were identified and acquainted through repeated observation. Subsequently, two persons observed the videos of the personality test and the pairwise experiment independently to record the behavioural variables. Finally, the records of the two observers were compared, and any inconsistent results were subsequently re-examined to ensure the reliability of the findings.”

Comment 15:

on page 3, lines: 133-135: " As there were no significant differences observed in behaviors between morning and afternoon, behavioral recordings were conducted once daily." There are studies that show fish activities and behaviour rates are different in the morning and afternoon. Did you any analysis to explain this result which would be also interesting. Also are there other studies finding the dame patterns? You need to elaborate this issue in the discussion part accordingly.

Response 15:

Thanks to your comment.  We referred to the result from Wong's (2013)– “There were no significant differences between morning and afternoon behaviours for each of the behavioural traits on the 3 days (ANOVA; P > 0·05 for all)”

Wong M Y L, Medina A, Uppaluri C, Arnold S, Seymour JR, Buston PM. 2013. Consistent Behavioural Traits and Behavioural Syndromes in Pairs of the False Clown Anemonefish Amphiprion ocellaris. Journal of Fish Biology. 83(1): 207–13. doi: 10.1111/jfb.12133.

Comment 16:

on page 3, line 135: "... with the initial 2 minutes allocated for adaptation..."

please explain why you use "adaptation" term? Why you do not use "acclimation" instead? The authors need to consider relevant and appropriate terminology in their methodological approaches.

Response 16:

Thanks to your comment. We have revised it.

Comment 17:

on  the page 3, line: 142 A picture or model of a companion fish is a conspecific, not a novel object so there is a methodological issue in this part of your study. Otherwise if you have any valid references to justify this issue. If not then you need to edit and revise it as using a conspecific model object.
Response 17:

We totally agree with your comment. We have revised it.

Line 141-146 – “two tests were used to assess the personality of each A. clarkii individual, specifically their reaction to two conspecific intruders. The first was an anemonefish model (of the same size as the examined individuals) introduced into the tank, and the second was a mirror image of itself (a mirror was placed in the tank). The latency time to first attack and the frequency of attacks on the model and mirror were observed and recorded within a 10-min period.”

Results:

in general the data were presented in an appropriate way but table 1 and figure 4 need revisions as mentioned bellow:

Comment 18:

On the page 5, line: 225: for the table showing behavioural ethograms please add any references that they also used the same description per each behaviour for the same or other species (also needs to be mentioned on the table) by adding another column on the table.

Response 18:

Thanks to your comment. We have added it

Comment 19:

on the page 6, line: 247: " ...appeasing behavior..." there is no indication of definition for this behaviour in the methods section, please provide more info about it in the text or in the table 1.

Response 19:

Thanks to your comment. We have added it in line 216-217—" indicative of submission and pacifying  opponent”

Comment 20:

on the page 6, lines: 247-248: ".... Additionally, some individuals did not clearly exhibit either of these two distinct personality traits. ..."  did you analyses further these individuals (other behavioural measurements/tests)?

Response 20:

Thanks to your comment. These individuals were the 7 out of 50 individuals who took the personality test whose results were inconsistent between the two tests. It was not possible to determine whether their personality was bold-aggressive or shy-submissive (gray circles in Figure 2) .

Comment 21:

on page 9, figure 4: please change figure number from "a" to "A" and the same for "b" as you referred them in the text accordingly.

Response 21:

We have revised them.

Comment 22:

Discussion and Conclusions:

there is no declaration of the limitations of the study. Also all the results need to be discussed from multiple angles and placed into context in separate paragraphs and /or well structures parts in the discussion section. More relevant references need to be included in the discussion section.

Response 22:

Thanks to your comment. We have added it in line 330-338.

Comment 23:

there is no conclusions included.

Response 23:

Thanks to your comment. We have added it.

Round 2

Reviewer 4 Report

Comments and Suggestions for Authors

The authors have addressed comments and updated /revised manuscript accordingly.

Comments on the Quality of English Language

The authors have addressed comments and updated /revised manuscript accordingly. Minor editing of English language required